# Health Perceptions in Relation to Child Health and Mortality in a Rural Context, Sierra Leone: A Mixed Method Study

**DOI:** 10.3390/ijerph18010308

**Published:** 2021-01-04

**Authors:** Camilla Midtgaard Eriksen, Monica Lauridsen Kujabi, Aminata Sulaiman Kanu, Gabriel Gulis

**Affiliations:** 1Unit for Health Promotion Research, University of Southern Denmark, 6700 Esbjerg, Denmark; camillamidtgaarderiksen@gmail.com; 2Global Health Section, Department of Public Health, University of Copenhagen, 1050 Copenhagen, Denmark; monica.kujabi@gmail.com; 3Faculty of Social Science, University of Makeni, Makeni, Sierra Leone; outreachofficer@masanga.dk

**Keywords:** health perceptions, child health, children under five, caregivers, Sierra Leone

## Abstract

Child survival and wellbeing remain a global health challenge despite vast development within the area and a significant decline in mortality rates of children under five years of age. This study investigates the perceived causes of ill health and childhood mortality in the context of five villages located in the Tonkolili district of Sierra Leone. Mixed method methodology was applied in this study consisting of both quantitative and qualitative data contribution. The quantitative part consisted of a household survey on child health, where 341 households, equivalent to 50.6% of the total number of households in the five villages, participated with a response rate of 100%. The qualitative part consisted of six semi structured interviews—one with a health care worker and five with mothers from each village. The main perceived reason for child morbidity was inadequate care of children related to personal hygiene of the child, hygiene and safety in the environment, in-sufficient nutrition, inadequate supervision and poor healthcare seeking behavior. Additionally, reasons given for disease included supernatural forces such as witchcraft. In relation to the survey, the perceived causes of child mortality for ill children in the villages were mainly malaria (33.6%), diarrhea (11.6%), pneumonia (8.6%), and unknown (26%). The observed symptoms of illness among children were fever (43.7%), cough and difficulty breathing (10.7%), frequent watery stool (10.3%) and no symptoms (20.3%). The perception of ill health in children was mainly associated with the parent’s ability to cater for the child’s physical needs, but also associated with external factors such as witchcraft and “God’s will”. In addition, biomedical causes for disease and supernatural causes for disease were seen to coexist.

## 1. Introduction

Despite global improvement, 5.3 million children died in 2018 alone before reaching their fifth birthday, mostly from preventable causes [1]. Many of the world’s children are unfortunately still left behind, with huge disparities existing between regions. Sub-Saharan Africa alone accounts for a staggering 2.8 million deaths among children under the age of five, 52% of the global under five mortality [1]. In-country disparities have also shown to affect chances of survival for children across the globe. Various studies show that the mortality in rural children is around 1.5 times higher than that in urban children, most likely reflecting the different conditions rural children have in accessing the education system and health services together with poorer living standards i.e., access to water, isolated housing, etc. [2,3]. Poorer children have 2.5 times higher mortality than children that are more financially well off, showing the unfortunate and strong link between child mortality and poverty [1,2,3]. Sierra Leone is sadly found at the lowermost end of the international statistics when it comes to child survival and is among the bottom five countries in the world, with a staggering child mortality rate of 105.07 per 1000 live births [4,5]. The leading causes of death in children under five years of age in Sierra Leone, are neonatal complications (29%), malaria (20%), pneumonia (12%) and diarrhoea (10%) [6]. Paradoxically, the world knows how to treat the majority of these diseases and prevent children from dying; the technical and medical solutions to the problems are well-documented [7]. 

However, the effectiveness of most medical interventions requires that the caregiver of the children utilise the health care system. Such utilisation may at large be influenced by community and individual perceptions and knowledge related to health and disease [8]. It is therefore of relevance to assess the caregiver’s perception of child health and morbidity, in order to understand some of the existing reasons for the high mortality rate in Sierra Leone. Health perceptions are widely studied across the African continent [8,9,10,11,12] however, regional perceptions can vary widely, and health perceptions need to be understood in their local context if health interventions are to be successful [8]. In Sierra Leone, little has been documented concerning community or caregiver’s perception on child health and mortality, and at a local level nothing was found on the subject in relation to Tonkolili District. This study was conducted in collaboration with Masanga Outreach, as part of the development and implementation of a Community Child Health Project (CCHP). Masanga Outreach, under the umbrella of Masanga Hospital, is a Non-Governmental Organisation (NGO) focusing on primary health in five local communities in the Tonkolili District in the Northern Province of Sierra Leone. Hence, this study aims to assess the caregiver’s perception of child health, morbidity and mortality, for the purpose of designing and implementing a context-appropriate child health intervention in the rural villages around Masanga Hospital.

## 2. Materials and Methods

### 2.1. Study Setting

The study was carried out during September 2019–March 2020 in five villages in Tonkolilli district, Sierra Leone; Masokoray, Makamba, Rosint, Marako and Rogbom Kamba. The villages ranged in size from 32 to 223 households. The villages belong to the catchment area of Masanga Hospital. They are all considered rural in the Sierra Leonean context, with 16–25 km distance to the nearest city, Magburaka.

### 2.2. Study Design and Data Collection

#### 2.2.1. Household Survey 

The study population was chosen in relation to the CCHP, prior to the survey and research aim of the presented study being set up. Households were chosen as the response unit of the survey due to the often-polygamous family structure in the five villages. A household in the survey was categorised as the immediate family: the father, his wife(s) and the children whom they are economically responsible for. Inclusion criteria were households with one or more child(ren) under the age of five. All children under the age of five were included in the survey, as long as the family was considered the main responsible caregiver. Only women were chosen as respondents because they are most often the primary caretaker of the children [13]. One woman would give a response on behalf of herself and her spouse, and in polygamous families, on behalf of her fellow wives as well, if they themselves were not present during the interview.

With the limited resources available, the nature of and reasons for data collection and the absence of valid demographic data, a randomised collection was not achievable, and convenience sampling was deemed necessary. Initially, the nearest house at the right-hand side to the chief’s house was identified and selected as the first house to assess for participant eligibility in each village. Every second household was chosen for participation to secure some form of randomization. Households without children below 5 years of age were excluded. The data was collected from the 28 October to the 15 November 2019. The response rate for the survey was 100%. The minimum target sample size was 245 households. This was calculated using the estimate of total households in the villages. To reflect maximal uncertainty, calculations were based on an estimated 50% prevalence of the main outcome (experience of child mortality in the household), a precision of 5% and a 95% CI. However, an oversampling of the calculated sample size of 245 household was chosen, in compensation for the lack of randomisation in the sampling method. The target for minimum sample size was therefore 50% of the households, in order to secure a sample that was reflective of the villages. This was further applied to each village so that every village was represented according to its size. A total of 341 households were included in the survey corresponding to 50.6% of the total number of households. 

The survey was interviewer administered due to the large proportion of illiterate people in the study population. There were three interviewers, all local staff from Masanga Outreach. All data collectors received a thorough introduction to the questionnaire and an in-depth training on data collection. The questionnaire was written in English because this was the language spoken between the primary investigator and the local data collectors. The questionnaire was not further translated into the local language Temne. However, the interview was performed in Temne between the interviewer and the participants. 

#### 2.2.2. Semi-Structured Interviews

A total of six interviews were collected, in the period of 24 January–7 February 2020. One interview was done with a health care professional and the remaining interviewees were mothers from the villages included in the survey (focus group interviews), all with children under the age of five. All informants were anonymised to respect their privacy and ensure confidentiality. Selection criteria for the women in villages was that they had a current child under the age of five. Every village was represented with an informant. To obtain several perspectives it was attempted to find women of different ages, with different educational backgrounds, different occupations and different family structures. Access was made possible through a local Community Health Committee. The interviews were based on an interview guide. The interviews were all first recorded as an audio file, and afterwards transcribed into words. All interviews were transcribed by the primary investigator. The transcripts were anonymised to secure confidentiality. The transcript aimed to be as true to the wording of the informants as possible. However, due to what is often referred to as African Pidgin English, where the phrasing is sometimes very different from standard English, statements were in some cases transcribed into a more formal language to make a more coherent text [14].

### 2.3. Data Analysis 

The data set for further use was created using Excel, where all the survey responses were typed in and coded manually from the paper survey responses. The data was all typed in by the primary investigator, then reassessed for flaws and missing values. The data was not assessed externally. The open-ended questions were separated into common themes then further coded into appropriate categories. The coded data was made into a STATA-dataset using StatTransfer (StatTransfer, Seatlle, WA, USA). Further data cleaning and statistical analysis was done using STATA 15 (StataNordic, Stockholm, Sweden). The statistical method used for analysing the survey data, was descriptive univariate analysis consisting of frequency distribution, point prevalence and central tendencies in the form of means. The data and informed consent form were stored in a locked file cabinet in the Masanga Outreach office, except in cases where the investigator was legally obligated to report specific incidents. After completion of the study, the questionnaires were maculated and destroyed.

The interviews were audio recorded and transcribed using NVivo 12 (QSR International, London, UK). After termination of the study, the recording and audio file was deleted. The written format of the interview was assessed and coded into several categories emerging from the material for further analysis. Informed consent was obtained from all participants before the beginning of the interview. 

## 3. Results

### 3.1. Socio-Demographic Characteristics

The number and distribution of participants by village (Figure 1) and marital structure (Figure 2) are presented in following Figure 1 and Figure 2.

Monogamous marriage/s were most common with 69.8% of the households, while 30.2% were polygamous households.

Table 1 Shows the distribution of the households in relation to size and content.

The mean of people in each household was 6.62 (95% CI 6.23–6.95) and the mean of children under the age of five in each household was 1.90 (95% CI 1.78–1.99). The mean of wives per household was 1.40 (95% CI 1.31–1.46). 

The socio-demographic characteristics are presented in Table 2 for males and females and for each of the number of wives, if several to one husband. 

The total of 100% is always calculated by columns. Among study participants 32.1% attended schooling while 67% received no formal schooling. The level of illiteracy was, nevertheless 77.5%. A total of 95.5% of all the participants earned an income and 61.5%, across gender, worked in gold mining, 18% worked in farming, 13% were traders, 5.8% worked in other areas, and 3.3% were unemployed.

### 3.2. Perceived Causes of Child Morbidity 

When it comes to immediate cause of illness, the participants in the survey were asked an open question to name all the reasons they could think of that caused children to fall ill. Answers were organized in categories and are presented as absolute numbers in Figure 3 (proximal causes) and Figure 4 (distal causes).

The high number of “other causes” is due to inability to define the cause.

Multiple answers were allowed on distal causes of diseases and the two most common reasons for disease were found to be lack of food and hygiene with 66% and 62.5%, respectively. 43.4% mentioned “not taking good care of the child”, which is defined and further elaborated in Table 3.

The variable ‘cold’ was a combination of a cold environment and children feeling cold or not being protected by enough clothes which was mentioned by 32.3% of the respondents. Moreover, ‘supernatural means’ is stated by 18.2% and is seen as a combination of both evil spirits and witchcraft. 

Identified from the semi-structured interviews, three main causes of child morbidity existed, namely: God’s will, not taking care of the child and supernatural means. When it comes to specific diseases, the perceived reasons for morbidity however varied (see Table 4).

Malaria is widely believed to be caused by oranges specifically, followed by sugary food, e.g., candy, soda and honey. The belief is that the oranges and sweet foods will make the blood glucose increase, and that an imbalance in the body will cause malaria. Interestingly, this belief weighed much more heavily than the belief that mosquitoes cause malaria. However, for many, both causes were simultaneously acknowledged or believed. 

When discussing diarrhoea, it was evident that informants believed that many causes were related to nutrition. Even though contaminated drinking water was mentioned, it did not weigh heavily among informant perceptions. Rather, certain types of food were seen as the main cause, often in combination with breastmilk—bad breast was an often-mentioned cause of diarrhoea (for further definition of “bad breast”, see Box 1). 

Box 1Definition of “*Bad Breast*”.Bad breast is defined as being ‘contaminated breastmilk in a lactating mother’. The contamination can be caused in several ways: 
Stagnated breastmilk in the breastPoor hygiene of the breastMenstruation periodIntercourse with spouse in breastfeeding period Some types of food, for example eggs


Pneumonia was interesting because only three causes of pneumonia were mentioned: exposing the child to a cold or moist environment and inhaling dust. 

Of further interest, Box 2 shows some answers of the informants when asked how they identified a supernatural cause of a disease, compared to a biomedical cause. 

Box 2Ways to identify supernatural causes for disease.
“The way the child will behave. Some they will fall and become stiff. That will let them know that this is because of demons.”“If the child’s eyes cannot close. If the child is like, having all the white, where all you can see is the white, the black will be up there, then they should go to the herbalist.”“They believe its evil spirits that causes convulsions when the child convulses, they will have to rub the child with palm oil and then garlic and these other things because, they believe it’s an evil spirit that has come close to the child.”“If the child is very, very weak and cannot move a lot, they should go to the herbalist. Because she believes it’s the demon. It’s the devil”


### 3.3. Perceived Causes of Child Mortality

In relation to the experience of child mortality, a total of 31.9% of the women had experience of child mortality, 13.9% of the women had lost one and 10.3% had lost two children (see Table 5).

Apparently, the first wives were more experienced with childbirth and child mortality issues.

Table 6 Shows the distribution of the age confirmed cases of child mortality below five years of age.

This shows that 89.9% of the child mortality happened before the age of two. 22.1% were neonatal deaths. The causes of child mortality in the villages relied solely on self-reported perception from the mothers included in the survey. The most commonly reported cause of death in children below the age of five was malaria with 33.6% of the reported cases, followed by diarrhoea at 11.6%, pneumonia at 8.6% and finally malnutrition with 4.6%. Evil spirits and witchcraft accounted for 5.4% of the deaths in children under five in the reported cases. The most common symptom reported was high fever which was seen in 43.7% of the cases. Cough and difficulties breathing were found in 10.7%, frequent watery stool or vomiting was found in 11.6% of the cases, convulsions were found in 9% of the cases and rapid weight loss and weak, thin children were reported in 4.4% of the cases. However, a substantial number of the women, 26%, reported not to know the cause of death of their child and 20.3% claimed that their children had no symptoms prior to passing away.

## 4. Discussion

In relation to the survey and the interviews, a phrase that repeatedly occurred was taking good care of the child. Hence, a common perception was, that in order for children to stay healthy, you have to take good care of them. While conducting the survey it became evident that taking good care of a child was associated with personal hygiene of the child, hygiene and safety of the environment, sufficient nutrition, adequate supervision and proper healthcare seeking behaviour. One separate study from Nigeria had similar results. The mothers believed that childhood pneumonia resulted partly from carelessness of the mother, which was associated with similar aspects such as not taking good care of the child (9). While seeking a more in-depth explanation of how and why certain aspects related to taking good care affect the health of the child, informants explained that some of these, (cold, special types of food, dirt, etc.) “does not fit the system of the child” as if exposing the child to any of these factors is found to somehow disturb a balance in the body, and this will subsequently lead to disease. An example of this is the way oranges are seen to disrupt the blood glucose and eventually lead to malaria. All of the mentioned aspects related to taking good care of a child are more or less related to the physical aspects of human wellbeing. 

In addition to perceived reasons for child mortality and morbidity, it became evident both from the survey and the interviews that biomedical reasons for disease and supernatural reasons coexist. This was evident in the survey where the participants were asked to name all the possible reasons they could think of for disease in children. Here, supernatural causes for disease such as witchcraft and evil spirits stand side by side with natural cause like contaminated drinking water, vector born malaria and lack of nutrients (poor or imbalanced nutrition). Hence, child health seems to a certain degree to be something that is influenceable by external factors. These factors can be affected by the caretaker’s behaviour towards the child, related to not taking good care, and to some degree is seen as being up to higher powers, such as witchcraft or God. 

One study from Burkina Faso found a similar cultural phenomenon regarding the co-existence of beliefs, where mothers reported both believing in malaria as being vector born and transmitted through mosquitoes, as well as caused by partially unrelated factors such as heavy oil consumption, work related fatigue, insufficient sleep, and exposure to sunlight. They also found that consumption of sweet fruits was seen as a cause of malaria [10]. This perception that fruit can cause malaria, is not something limited to Sierra Leone, or West Africa alone. A study from Mozambique also found a connection between fruit consumption and malaria transmission, though they reported mosquito transmission to be the main believed cause of malaria [11]. 

The perception that breastmilk can be contaminated or dirty was equally found in a study from South Africa, where germs and contaminated drinking water as well as witchcraft were identified to cause diarrhoea [12].

This co-existence of both the biomedical (natural and scientific) and the traditional (cultural beliefs and superstitions) paradigms seems to go hand in hand with the use of both facility-based medical treatment and traditionalists (herbalists, healers and traditional health experts). The use of traditional medicine across Africa is quite substantial with up to 80% use across the region and is therefore not a phenomenon limited to Sierra Leone alone [15]. In our investigation, biomedical treatment was seen to be effective in several areas, for example, in treating malaria, diarrhoea and pneumonia. The herbalist, however, was seen to be able to take care of the same, together with conditions perceived as being only caused by supernatural causes. Neurological symptoms such as convulsions, stiffness and severe fatigue were perceived to be rooted in supernatural causes alone. When asked who would best treat the child when sick from witchcraft, the agreed answer among the informants was the herbalist. In relation to identification of what was caused by witchcraft and what was a natural cause of disease, respondents noted that either elders or herbalists would be able to identify the nature of the cause. This might lead to health care seeking behaviour where the herbalist is sought before the healthcare facilities. 

However, in the survey, only 0.3% stated that the first person they seek medical advice from is the herbalist. This seemed inconsistent with the interviews and participant observations and also differs from other studies in Sierra Leone on the use of herbal medicine, where over half reported using herbal medicine (15). Some of the informants seemed to vary in their responses when it came to the use of herbal medicine. The reason there seemed to be a hesitancy talking about the use of herbal medicine and supernatural causes of disease might be due to the fact that the primary investigator, the translator, and the data collectors all represented the biomedical health system. This reluctance to talk to health professionals about their use of herbal medicine also became evident when the informants were asked if they would tell the health care professional at the hospital about their use of traditional remedies. Five mothers said they definitely would not. This tendency could potentially result in skewed responses and explain the low reporting to the biomedical or modern medical specialist when it comes to use of herbalists.

One reason for first seeking help at the herbalist was because of *the system*. What exactly informants meant by “system” was not specified. However, it could be hypothesized that this includes lack of access to the formal medical system due to distance, costs and trust issues. The use of herbal remedies also seemed to be because informants claimed they saw an effect from the herbs and therefore it made sense for people to use them. All the informants, furthermore, stated that they trusted the herbalist and her/his knowledge. Furthermore, the health care worker specified in the interview “*they trust them; that is the first point before seeking medical care*”. Two studies from South Africa and Kenya also stated that traditional healers were used because they were trusted and respected in the communities [16]. Another very central aspect when investigating the reasons for using the herbalist is the lower cost of herbal medicine and traditional remedies compared to a hospital visit. Bakshi et al. reported the same finding and reported a higher dependence in rural areas compared to urban [16].

In the current survey, 94.1% stated that the nurse or the doctor would be the best person to help their child when ill. Several informants in the interview confirmed this. When the health care worker in the interview was asked if she believed that the hospital would be used more than herbalists if the people in the villages had the time, the money and the social support, she replied “*they would go to the hospital, I am sure of that*”. 

Another trend noted in this study and found in other studies is the fact that people use the herbalist first, and only seek medical attention if the condition does not improve, or it worsens despite the use of traditional remedies. Bakshia et al. found that people in Sierra Leone, especially in rural settings, only sought facility-based treatment for their children if the condition severely worsened [16,17]. This can lead to severe delay in treatment and has potentially fatal consequences for children’s health and likelihood for survival when faced with illness. Another concern regarding the safety of the children is as expressed by the nutritionist and the nurse in charge of the pediatric ward: the toxicity of the herbs used for treatment. Ranasinghe et al. points to the fact that evidence on this area is limited and also inconsistent [15]. Delay in diagnosis and treatment caused by fear, misperceptions and lack of trust also played a significant role also in the case of the Ebola epidemic in Sierra leone in 2014 [18]. 

### Limitations of the Study

One of the limitations of the study is the fact that the primary investigator single-handedly performed the management and analysis of the data. However, as a positive trade-off, this allowed for more consistent interpretation and coding of primary data as well as increased consistency in analysis and interpretation stages.

Secondly, because no accurate geographic and census data was available on how many households there were in what specific compound and no data was available on how many individuals were residents in the villages, a simple random sampling technique was not possible. Furthermore, due to the nature of the survey and the initial purpose of the data collection (namely, to investigate child health in relation to the Community Child Health project in Masanga Outreach), a cluster sampling method was also not appropriate. Thus, a randomised collection was not achievable, and a convenience sampling was deemed necessary. It was decided that an oversampling of the calculated sample size of 254 household would compensate (e.g., if not representativeness, then at least a reflectiveness of the target population was secured). The target for minimum sample size was therefore raised to be 50% of all the households. 

Thirdly, an information bias that was suspected in the study was social desirability response bias. This is suspected because of the observed inconsistency between the survey answers and the interviews. If only taking the interview responses and participant observations into consideration, the survey seems to be a more positive representation of life for children in the villages. Therefore, there is a risk that the survey underrepresents or diminishes the actual challenges and barriers the informant communities face.

Fourthly, another limitation is language; especially in the case of qualitative interviews, lack of detail translation/re-translation and therefore detailed semantic analysis could lead to bias. However, the use of spoken interviews and surveys using local languages and face to face interactions helped compensate for language difference biases. 

Fifthly, only females and mothers were the primary target population for data collection. It might be important to include fathers and extended family responses in future studies (aunts, uncles, cousins, grandparents, and even caregiver friends).

Lastly, other limitations, as usual, include time, funding, staff, accessibility and other support constraints normative of many research endeavours in most fields. Nevertheless, with a sample set of 50% of the population and 100% response rate, these normative constraints were adequately compensated for, especially thanks to the positive support of local communities, leaders, and healthcare personnel and institutions.

## 5. Conclusions

A mixed-method research project was conducted in order to assess perceived healthcare problems, causes, and treatments for children five years old and younger among impoverished gold mining and farming villages in the Tonkolili district of Sierra Leone. 50% of households were sampled with a 100% response rate. The main perceived cause of ill health was lack of care, expressed as “not taking good care of the child”. This was related to poor hygiene practices, unsafe environment, insufficient nutrition, inadequate supervision and poor healthcare seeking behaviour. The perception of ill health in children was to a high degree associated with the parent’s ability to cater for the child’s physical needs. A more in-depth explanation of why and how these aspects affect the child was believed to be because these factors could upset the system of the child, which was seen to have a kind of inner balance. In relation to perceived causes of child mortality, biomedical causes of disease such as malaria, pneumonia and diarrhoea, together with supernatural causes for disease such as witchcraft, evil spirits and God’s will, were seen to coexist. A clearly defined line between the two did not seem to exist and the biomedical causes of disease were seen to be caused either because the children were not taken care of, or because of supernatural causes. Children’s ill health and mortality was thereby seen both as an element that could be influenced by the parent and their actions (or lack thereof), and simultaneously something that was out of the caretaker’s hands, and up to other people such as those able to bewitch, or up to the will of God. 

One of the key findings of the study is that of traditional perceived causes of malaria as being related to diet and other factors not causally related to the actual parasite and mosquito transmission—again demonstrating how important and strong are the traditional beliefs of the local population. This also relates to the often-preferred use of local herbalists and medical healers instead of modern specialists, clinics and other facilities. The preferred first use of herbalists and elders in many cases may be economic. Clinics and modern treatment are more expensive. However, it also may be related to traditional beliefs and practices as well as other socio-cultural aspects, such as trust. Locals trust and respect the local elders, healers and herbalists. This is powerful and influential in regard to parental choices and actions. In many cases, only if conditions for the children worsen is professional modern treatment sought. By then, successful treatment may be too late. These issues need to be taken into account while planning future research, health literacy programs, disease prevention practices, disease treatment practices, treatment options, and overall healthcare planning in Sierra Leone and similar settings.

## Figures and Tables

**Figure 1 ijerph-18-00308-f001:**
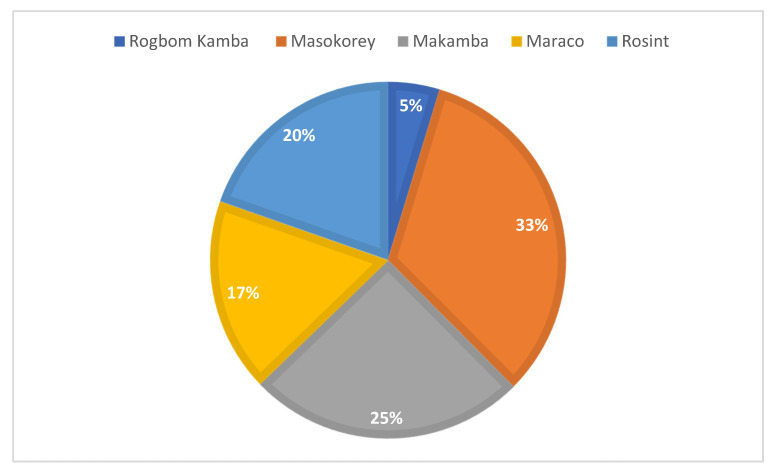
Distribution of respondents by villages.

**Figure 2 ijerph-18-00308-f002:**
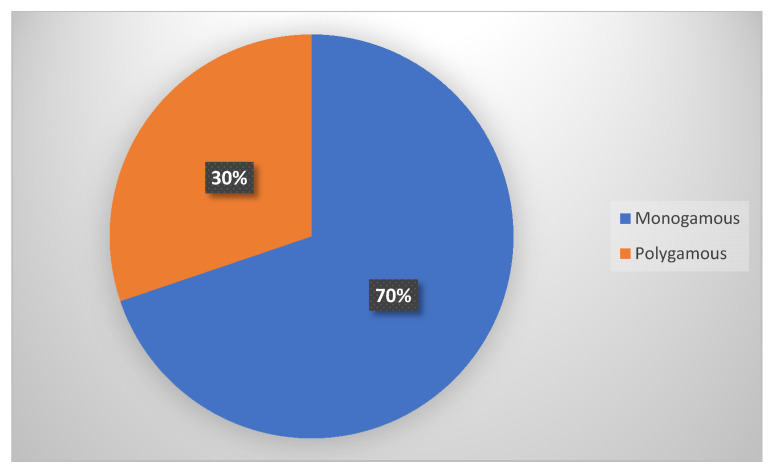
Distribution of respondent by family status.

**Figure 3 ijerph-18-00308-f003:**
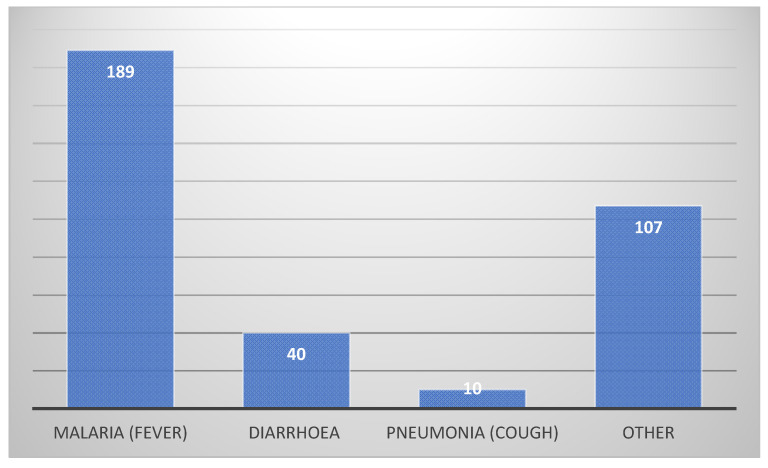
Proximal causes of childhood disease.

**Figure 4 ijerph-18-00308-f004:**
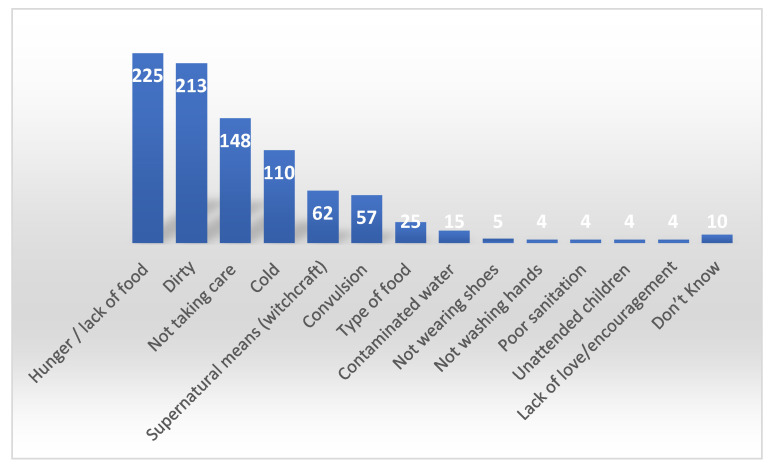
Distal causes of childhood disease.

**Table 1 ijerph-18-00308-t001:** Central Tendencies in Household Distribution.

Variable	Mean	Std. Err.	95% Conf. Interval
Number of people in households	6.62	0.16	6.23–6.95
Number of wives per husband	1.40	0.03	1.31–1.46
Number of children <5 in households	1.90	0.05	1.78–1.99
Number of live births per woman	3.78	0.11	3.34 to 4.26

**Table 2 ijerph-18-00308-t002:** Socio-demographic characteristics by gender and number of wives.

Variable	Male *n* (%)	Female *n* (%)	Wife 1 *n* (%)	Wife 2 *n* (%)	Wife 3 *n* (%)	Wife 4 *n* (%)	Total *n* (%)
Age in years							
15–25	36 (10.6)	161 (33.8)	131 (38.4)	23 (21.7)	7 (28)	0 (0)	197 (24.1)
26–35	127 (37.2)	167 (35.1)	124 (36.4)	34 (32.1)	7 (28)	2 (50)	294 (36)
36–45	87 (25.5)	53 (11.1)	38 (11.1)	11 (10.4)	3 (12)	1 (25)	140 (17.1)
46–55	28 (8.2)	12 (2.5)	8 (2.4)	4 (3.4)	0 (0)	0 (0)	40 (4.9)
>56	9 (2.6)	0 (0)	0 (0)	0 (0)	0 (0)	0 (0)	9 (1.1)
Missing	54 (15.8)	83 (17.4)	40 (11.7)	34 (32)	8 (32)	1 (25)	137 (16.8)
Level of school							
No formal schooling	215(63.7)	325(70.8)	231(68.1)	74(79.6)	16 (69.6)	4 (100)	540(67)
Primary school	27 (8)	48 (10.5)	38 (11.2)	7 (7.5)	3 (13)	0 (0)	75 (9.4)
JSS (middle school)	39 (11.6)	58 (12.6)	46 (13.6)	9 (9.7)	3 (13)	0 (0)	97 (12.2)
SSS (high school)	47 (13.9)	25 (5.4)	21 (6.2)	3 (3.2)	1 (4.3)	0 (0)	72 (9)
Voc. Com. Teacher	9 (2.6)	3 (0.7)	3 (0.9)	0 (0)	0 (0)	0 (0)	12 (1.5)
Higher education	0 (0)	0 (0)	0 (0)	0 (0)	0 (0)	0 (0)	0 (0)
Ability to read							
Literate	99 (29.4)	80 (17.4)	64 (18.8)	11 (11.8)	4 (17.4)	1 (33.3)	179 (22.5)
Illiterate	238 (70.6)	379 (82.6)	276 (81.2)	82 (88.2)	19 (82.6)	2 (66.6)	617 (77.5)
Earning an income							
Yes	328 (98)	444 (94.9)	315 (93.8)	100 (99)	25 (78.1)	4 (100)	772 (95.5)
No	7 (2)	29 (6.1)	21 (6.3)	1 (1)	7 (21.9)	0 (0)	36 (4.5)
Occupation ^1^							
Mining	223 (65.4)	279 (58.7)	194 (56.9)	65 (61.9)	18 (72)	2 (50)	502 (61.5)
Farming	69 (20.2)	78 (16.4)	57 (16.7)	20 (19)	1 (4)	0 (0)	147 (18)
Trading	20 (5.9)	86 (18.1)	63 (18.5)	16 (15.2)	5 (20)	2 (50)	106 (13)
Other	41 (12)	6 (1.3)	4 (7.2)	2 (1.9)	0 (0)	0 (0)	47 (5.8)
Unemployed	6 (1.8)	21 (4.4)	20 (5.9)	1 (1)	0 (0)	0 (0)	27 (3.3)

^1^: one person could hold several occupations.

**Table 3 ijerph-18-00308-t003:** (Not) taking good care of children (defined from the semi-structured interview and field notes).

Taking Good Care of Children	Not Taking Good Care
Ensure personal hygiene of the child Providing good and enough nutrition/feeding the child on demand Seeking health care when needed and having good medical compliance Supervision of adult Protection from outside dangers/making sure your child wears clothes and shoes	Neglected hygiene of the child Not feeding on demand/not providing enough nutrition/ giving the child food that is perceived as harmful Not seeking medical advice/letting the condition worsen before visiting a health care facility Leaving children unattended and unsupervised, often for long hours Not protecting the child from sharp objects/cold environments

**Table 4 ijerph-18-00308-t004:** Perceived causes for child morbidity.

Malaria	Diarrhoea	Pneumonia
Mosquitoes/mosquito bites Not sleeping under a bed net Exposing the child to a cold environment Eating too many oranges Eating too many sugary foods: sodas and candy	Contaminated drinking water Contaminated food Unclean environment and improper use of latrines Having intercourse while still breastfeeding Breastfeeding while having your period Bad Breast; contaminated breastmilk Some type of nutrition: some beans, sweet potatoes, unripe mangoes, too many eggs, too many bananas.	Exposing the child to a cold environment Exposing the child to a moist environment Inhaling (contaminated) dust from the environment

**Table 5 ijerph-18-00308-t005:** Birth characteristics and experience of child mortality.

Variable	Wife 1 *n* (%)	Wife 2 *n* (%)	Wife 3 *n* (%)	Wife 4 *n* (%)	Total *n* (%)
Age at first birth					
<12	3 (0.9)	0 (0)	2 (8)	0 (0)	5 (1,1)
13–15	203 (60)	43 (41)	11 (44)	2 (20)	259 (54.5)
16–18	69 (20.2)	27 (25.7)	6 (24)	2 (50)	104 (21.9)
19–21	21 (6.2)	11 (10.5)	4 (16)	0 (0)	36 (7.6)
22–24	1 (0.3)	0 (0)	0 (0)	0 (0)	1 (0.2)
>25	0 (0)	0 (0)	0 (0)	0 (0)	0 (0)
Missing	44 (12.9)	24 (23)	2 (8)	0 (0)	70 (14.7)
Number of live births					
1	52 (15.3)	16 (15)	2 (8)	1 (25)	71 (14.9)
2	63 (18.5)	27 (25.5)	7 (28)	0 (0)	97 (20.4)
3	55 (16.1)	15 (14.2)	4 (16)	2 (50)	76 (16)
4	54 (16.4)	12 (11.3)	3 (12)	1 (25)	70 (14.7)
5	35 (10.3)	9 (8.5)	2 (8)	0 (0)	46 (9.7)
>6	75 (22)	19 (17.9)	5 (20)	0 (0)	99 (20.8)
Experienced Child Mortality					
Yes	117 (34.3)	28 (26.9)	5 (20)	1 (25)	151 (31.9)
No	221 (64.8)	75 (72.1)	19 (76)	3 (75)	318 (67.1)
Number of deceased children <5					
1	52 (15.3)	10 (9.7)	3 (12.5)	1 (25)	66 (13.9)
2	35 (10.3)	13 (12.6)	1 (4.2)	0 (0)	49 (10.3)
3	19 (5.6)	3 (2.9)	0 (0)	0 (0)	22 (4.6)
4	4 (1.2)	1 (1)	0 (0)	0 (0)	5 (1.1)
5	4 (1.2)	1 (1)	0 (0)	0 (0)	5 (1.1)
>6	7 (2.1)	0 (0)	1 (4.2)	0 (0)	8 (1.7)

**Table 6 ijerph-18-00308-t006:** Perceived Causes of Mortality in Deceased Children.

Variable		Total *n* (%)
Age in month at time of death	<1 month	61 (22.1)
	2–6 months	63 (22.8)
	7–12 months	70 (25.4)
	13–24 months	54 (19.5)
	25–36 months	11 (4)
	37–48 months	9 (3.3)
	49–59 months	8 (2.3)
Place of death	In the village	50 (40.7)
	At the PHU or hospital	41 (33.3)
	Some in the village, some in the hospital	32 (26)
Perceived cause of death	Don’t know	81 (26)
	Diarrhoea	35 (11.6)
	Malaria	101 (33.6)
	Lower Respiratory Infection	26 (8.6)
	Malnutrition	14 (4.6)
	Skin infection	4 (1.3)
	Evil spirits	8 (2.7)
	Witchcraft	8 (2.7)
	Gods will	3 (1)
	Other *	16 (5.3)
Symptoms	No symptoms	61 (20.3)
	Frequent watery stools/vomiting	31 (10.3)
	High fever	131 (43.7)
	Cough	8 (2.7)
	Difficulty breathing	24 (8)
	Unconsciousness	7 (2.3)
	Convulsions	27 (9)
	Rapid weight loss	8 (2.7)
	Weak thin child	5 (1.7)
	Other	12 (4)

## Data Availability

The data presented in this study are available on request from the corresponding author due to ethical and privacy restriction reasons.

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
