# Peer review of "Health Perceptions in Relation to Child Health and Mortality in a Rural Context, Sierra Leone: A Mixed Method Study"

_ijerph, 2021, doi:10.3390/ijerph18010308_

Round 1
Reviewer 1 Report
The idea behind this study is very interesting and could bring promising development for future research.
However, I have some remarks that the authors could use to improve the content of their paper:
- The authors should mention the whole period of their study in paragraph between lines 71-74 as we got just the period of the interviews in line 111.
- Sierra Leone was among the first sources of highly infected regions by Ebola, but the authors did not take that into account as it is interesting to do so since the control measures which were followed against this epidemic suffered and were perturbed . I think this part deserves a little paragraph as there are good results in the following examples:
- Yamanis, T., Nolan, E., & Shepler, S. (2016). Fears and misperceptions of the Ebola response system during the 2014-2015 outbreak in Sierra Leone. PLoS neglected tropical diseases, 10(10), e0005077.
- Wigmore, R. (2015). Contextualising Ebola rumours from a political, historical and social perspective to understand people’s perceptions of Ebola and the responses to it. Ebola Response Anthropol Platf, 4. - The following reference could also be discussed as the authors did not give enough examples about their final statement in lines 31-32, please see:
- Ujah, Innocent AO, et al. "A review of perception and myth on causes of cholera infection in endemic areas of Nigeria." African journal of microbiology research 9.9 (2015): 557-564. - In order to help the reader to check your results, it would be very interesting to add some statistical simulations to this paper, like using pie charts and histograms, instead of tables only.
- The authors may also give their opinion or advice for the best management of corona in that region, based on their conclusions.
Author Response
- The authors should mention the whole period of their study in paragraph between lines 71-74 as we got just the period of the interviews in line 111. we added September 2019-March 2020 as study period on line 71.
- Sierra Leone was among the first sources of highly infected regions by Ebola, but the authors did not take that into account as it is interesting to do so since the control measures which were followed against this epidemic suffered and were perturbed . I think this part deserves a little paragraph as there are good results in the following examples:
- Yamanis, T., Nolan, E., & Shepler, S. (2016). Fears and misperceptions of the Ebola response system during the 2014-2015 outbreak in Sierra Leone. PLoS neglected tropical diseases, 10(10), e0005077.
- Wigmore, R. (2015). Contextualising Ebola rumours from a political, historical and social perspective to understand people’s perceptions of Ebola and the responses to it. Ebola Response Anthropol Platf, 4. We understand your point, however adding the Ebola issue to the paper would slightly modify its focus. It was not part of planned research protocol and as visible from results, Ebola was not enlisted by study participants as a major health issues. Therefore we decided not to add it into discussion. - The following reference could also be discussed as the authors did not give enough examples about their final statement in lines 31-32, please see:
- Ujah, Innocent AO, et al. "A review of perception and myth on causes of cholera infection in endemic areas of Nigeria." African journal of microbiology research 9.9 (2015): 557-564. Thank you, we added this reference into text and reference list - In order to help the reader to check your results, it would be very interesting to add some statistical simulations to this paper, like using pie charts and histograms, instead of tables only.Although we agree with reviewer, due to shortage of time and other work commitments in different countries of Africa at the time of this revision, we are unable to change the tables to figures. That would require longer time.
- The authors may also give their opinion or advice for the best management of corona in that region, based on their conclusions.This research was completed before Covid-19 erupted, therefore it would be irresponsible from us to make conclusions or recommendations toward control of Covid-19. Covid-19 is apparently very different issue as child mortality nad we believe it should be mixed into this manuscript.
Reviewer 2 Report
Review: Health Perceptions in Relation to Child Health and Mortality in a Rural Context, Sierra Leone: A Mixed Method Study
Overall: good study, data rich, relevant, useful, but the analysis, writing and paper must be improved. There are many details about the research and methodology that are missing but critically needed. See comments below.
Many of the following statements are comments, suggestions, and questions. The authors are at liberty to use, modify or ignore at their discretion. I may have misinterpreted some areas; erroneously assumed certain lines of logic in the methods, analysis and interpretations at some points; etc. Please ignore and forgive if this is the case.
Line 14: “… remain a significantly critical global health challenge…” Note: Word Choice - should avoid relatively ‘colloquial’ words and expressions such as “huge”. True, huge can mean “of great importance or seriousness” but is not necessarily the best for academic journals. Some readers may take the authors less seriously. However, you do not need to use overly complex language. Keep is simple, but accurate, precise, professional and ‘readable’ (i.e., readability yet professionalism will increase the power of your article and by extension, the research).
Line 15: Need to define “under five”. I assume it’s an age group, but not everyone will. “… decline in mortality rates for children under five years of age (under five)…”
Note: Need to edit the article for word choice, phrase choice, sentence structure, and overall academic English “readability”. The data and results are interesting and seem solid enough. The qualitative sample set seems relative small (6 respondents; or were they focus groups?), but I understand the limitations with sample size for semi-structured less formal but more intensive open interviews (e.g., finding willing and appropriate respondents; building trust and rapport; coding and interpreting the ‘textual’ and more nuanced as well as emotive data to compare with the quantitative structured/questionnaire method, etc.). However, although the sample sets are often smaller, the results are generally richer, more nuanced, and more accurate. By the way, need to describe what type of quantitative method was used in abstract: structured questionnaire with Likert scale based questions administered verbally by a research team member; ‘checkbox’-based questionnaire (choose from a structured list; and, is there a limited number of choices to make, are they ranked, etc.); multiple choice; dichotomous questions… No need to be overly descriptive, but let the readers know basically that the “survey” was some form of quantifiable structured instrument delivered verbally by research team member due to high illiteracy rates. This also lets the readers know the area is highly illiterate and this is an important factor in child healthcare, or healthcare in general.
Suggestions for further abstract revision are as follows:
“…. The perceived causes for ill health and childhood mortality were investigated in this study. The sample area included five villages located in the Tonkolili district of Sierra Leone. A mixed method methodology was applied in consisting of both quantitative and qualitative approaches. The quantitative method consisted of household surveys for 341 households (50.6% of the total number of households in the five villages; 100% response rate). The qualitative method consisted of six semi-structured interviews - one with a health care worker and five with mothers from each village. The main perceived reasons for child morbidity were inadequate care of children related to personal hygiene of the child, hygiene and safety of the environment, insufficient nutrition, inadequate supervision, and poor healthcare seeking behavior. Additionally, reasons for disease included supernatural forces such as witchcraft. In relation to the survey, the perceived causes of child mortality for ill children in the villages were primarily malaria (33.6%), diarrhea (11.6%), pneumonia (8.6%), and unknown (26%). The observed symptoms of illness among children were fever (43.7%), cough and difficulty breathing (10.7%), frequent watery stool (10.3%) and no symptoms (20.3%). The perception of ill health in children was mainly associated with parental ability to cater to a child’s physical needs, but also associated with external supernatural and religious factors such as witchcraft and “God’s will”. In addition, biomedical causes for disease and supernatural causes for disease were seen to coexist.”
Lines 53-55: care gives perceptions and knowledge related to health and disease.
Lines 58-59” “… health perceptions need to be understood in its local context…” This is an astute point often neglected or not well understood by larger international organizations (e.g., WHO) who (forgive the pun) often generalize and take a ‘one size fits all’ view and approach to problem identification, research, and solution development.
Line 83: Even though women are the primary caregivers of the children, why exclude men? I would think the male perceptions would be an interesting comparison, more than marginally important to overall cultural perceptions, and there is always the possibility the males (at least to certain degree) may have significant influence over the female perceptions.
Lines 88-102: Sampling actually quite good with the 100% response rate and every second household sampled (341 households were sampled – 96 households more than the initial target of 245… somewhere around a 40% increase - impressive). No need to worry about ‘random sampling’ in such cases. Random sampling is more or less a representative bootstrapping technique for small sample sizes in large populations. Also, targeted sampling is often more effective for many types of studies.
Note: Even though ‘informed consent forms’ are mentioned to have been stored (Line 133), and Lines 139 and 140 states informed consent was obtain, at some earlier point, the authors may want to mention if the research was IRB approved, informed consent was practiced, ethical standards were adhered to (eg, Belmont Report), etc.
Line 107: Would be more professional to have translated the questionnaire into Temne as well. As it stands, it seems like the research implementation was designed to pamper the primary investigator. Also, some nuances are often obscured. For example, health investigators in an Indonesian study (I’ll keep it anonymous) asked certain ethnic groups about malaria cases (using the national language term ‘malaria’). They determined there were up to 90% infection rates in many areas. However, the fact was that malaria to many groups simply meant ‘anything with a fever’ such as intimated in Table 4 of this paper. Thus, anything with a fever could have been a wide range of viral, bacterial, parasitic, etc. infections. Later studies with local language considerations and blood testing, etc. found the actual malarial rates were much lower; in some cases errors up to 90%. Thus, it makes it very problematic and very expensive to treat actual problems appropriately and effectively by health organizations.
Additionally, was the full questionnaire/instrument included in this paper? I didn’t see it as an appendix or attachment. It would be helpful to include if not included.
Lines 110-124: I understand the need for a diverse group to be interviewed even though only six interviews were conducted. This allows for variability. However, some reviewers may frown upon this approach. I have no problem with it, however. The intention is to accommodate diversity and understand a more accurate and representative range of variation.
Line 148 Table 2: OK, but may want to add max/min, range, standard deviations. Personally, a histogram of the actual family sizes would be easier and more readable and perhaps more meaningful.
Line 154 Table 3: This table is a little busy and confusing. The percentages need to be further explained. Percentages of what totals for example. As it is presented, the reader has to figure out this by actually doing the math his/herself. The table needs more clear explanation of data presentation. I would suggest simply providing a more simplified and clear table. Also, demographic data such as the large percentage of miners (both male and female) are relevant for inclusion in the abstract, introduction, etc. This should include the type of mining they do and the conditions as well (local industry gold mining versus heavy industry mineral mining…).
Line 166 Table 4: Needs clarifications and further explanation, especially the distal causes of disease categories percentages. Although the textbox explanations are useful (Lines 171 and 179), the data presentation, tables and explanations/elaborations can be better presented. The textboxes can be better designed. Nonetheless, it is good data, interesting results, relevant and useful.
Line 189 Textbox 3: Bad breast is an interesting topic. Further elaboration would be useful. This also reminds me, the authors need to discuss the current physical and social environments (related to pneumonia below) – especially anything extreme, unsanitary and toxic that may exist – from weather to unclean water to impure/polluted air. The authors also need to briefly discuss the current access to health information and healthcare personnel as well as facilities and treatment. Returning to the bad breast topic, how much information on breastfeeding and lactation is available? If available, do women resist changing their beliefs and practices and why?
Table 5: Again, needs clarifications and cleanup. For example, need to explain that percentages are based on column totals and need to explain what the final total column informs, etc. Also, why is Table 5 in embedded in the middle of Textbox 4?
Note: Because women often have children quite young (13-15: 60%; 16-18: 20% for the first wife; 13-15: 55%; 16-18: 22% for the totals), it would be expected that child morbidity/mortality rates would be quite high for the first pregnancy/child. This should be indicated in the abstract and introduction as well.
Note: Discussion of the nature and role of the local herbalists would be useful, although the authors do provide a useful summary in the discussion (Lines 247-. Also, are herbalists undermining other forms of healthcare initiatives or helping them? This seems to be suggested by the authors in Lines 258-259. This is interesting and an important point noted by the authors. I would encourage further analysis and discussion. The inconsistencies noted among the respondents compared to other studies and subsequent analysis and explanations provided by the authors stated in Lines 259-292 are good. This is useful information and a good analysis.
Line 201 Table 6: Good summary data. However, the table could be aligned or designed better. Center justification may not be the best way to present the information.
Lines 217-218: Mention “participant observation” which is standard anthropological method that differs from ‘surveys’ (questionnaire based structured interviews in this case) and ‘in-depth semi-structured face to face interviews’. Please add details to the participant observation aspect if this is indeed a third method employed in the study.
Lines 229-230: The coexistence of biomedical and supernatural explanations is an interesting topic. However, are the authors distinguishing the witchcraft and religious as separate or is religious belief included in the supernatural category? It seems primarily the witchcraft although God is mentioned in line 236. The discussion and comparisons are good (e.g., Burkina Faso, Mozambique, South Africa and fruit cause of malaria as well as contaminated/dirty breast milk).
Lines 231-232: The authors note that respondents were asked to list all possible reasons for disease. This (and much more detail) needs to be mentioned up front in the methods section.
Lines 214-292 Discussion: Overall, some excellent points are highlighted by the authors, especially concerning herbalists, formal medical facilities/personnel, trust (trusted people and traditions), economic considerations, consequences, and so forth. I would encourage more in-depth analysis and discussion, along with recommendations for possible solutions.
Note: A discussion of the methodology would be useful. That is, did the various methods synch, deviate significantly, deviate in certain areas, complement each other, undermine each other, etc. and why? Was this mixed method approach successful, partially successful, etc.? How could it be improved?
Lines 310-324 Conclusion: This is fairly brief and could be improved. Also, do the authors want to/need to make recommendations to help alleviate existing problems and recommendations to enhance future research? This would be useful in my opinion.
Lines 334-336 Acknowledgements: I’m a bit confused. Is this left over from original author guidelines?
Lines 338-387 References: This is somewhat ‘thin’. Although many of the research references and comparative study references are appropriate, a more thorough literature review may be in order.
Author Response
Thank you for your valuable comments! We will address them individually in following, referring to identifiers provided by you by black letters and our comment by blue color.
- Line 14 - we deleted the word "huge"
- Line 15 - thank you, corrected in text as you suggested
- Note -thank you for this note! We tried to specify as much as we can in limited time available
- Suggestions for Abstract...- we accepted and implemented most of your formulations, thank you
- Line 53-55 - Line 56 now, but here we left the original, as we believe it is about caregiver's perception indeed
- Lines 58-59 - thank you, full agreement
- Line 83 - interesting point indeed! The reason of focusing on mothers in our study was given by economic and cultural context; fathers are mostly out of household. However, based on your point we added a final sentence in Conclusion asking for this approach in future studies.
- Lines 88-102 and Note - thank you again. Yes, the Masanga Outreach project was approved and this research is part of it.
- Line 107 - we agree, but unfortunately due to lack of time and resources that was not possible.
- Line 110-124 - agree
- Lines 148, 154 and 166 concerning Tables 2, 3 and 4- we tried to add some explanatory text to tables. Some of them could be transformed to figures, but that would require longer time for revision due to other commitments of co-authors in different countries at the time of this revision
- Line 189 - we agree that it is a pity, but within the study we did not collect ore information about breastfeeding practices or specific information available. Due to this, we cannot add more information on this item.
- Table 5 - added a clarification sentence after table. Sorry for the mix-up with the text box; on my PC it is separated of the text box therefore it must be an editing problem only which will be solved during final proofs.
- Table 6 - we changed the justification
- Line 217-218 -now line 229 - we changed the formulation to avoid this misunderstanding
- Line 229-230 - considered together as supranatural
- Line 231-232 - it is mentioned now below the relevant table
- Discussion and Conclusions - thank you for motivating to provide ore recommendations. However, we want to stay rather neutral with these two parts as we believe the methodology of the study does not allow for more strong conclusions. This work was done as a master thesis, and you mention many points which would require more in depth information (bad breast), or methodological changes (translation of questionnaire to temme).
- Acknowledgement - yes, that is a reminder from the submission template; there is a factual acknowledgement included within submission process.
- References - based on recommendation of other reviewer we added one more reference now. The master thesis, which is a source of this manuscript, contains substantially more references, but again, we aimed to make the manuscript short and the number of references reflects this.
Reviewer 3 Report
Thank you for the opportunity to review the manuscript titled, “Health perceptions in relation to child health and mortality in a rural context, Sierra Leone: a mixed methods study.” In this study, the authors aimed to investigate the perceived causes for ill health and childhood mortality in context of five villages in a single district of Sierra Leone.
Quantitative data, in the form of household surveys, was collected from 341 households (50% of all possible households in the villages).
Qualitative data, in the form of semi-structured interviews, were held with one healthcare worker and five mothers from each village. This was an ambitious study, both in term of design and study location.
The use of mixed methods gleaned insight (and fascinating ones at that), which would not have been possible using either method alone.
That said, I do have some comments that the authors may wish to consider. 、1. Lines 72-73: The authors write that the “villages ranged in size from 32 to 223 households.” Was there consideration to village size when analyzing data on perceived causes of ill health? 、
2. Similarly, was there consideration to polygamous vs monogamous households? (Lines 79-86).
3. Please provide a few words about how you were able to attain a survey response rate of 100% (Line 93).
4. Lines 106-9: Thank you for the clarification on the administration of the surveys and the language used. Please add a few words as to how translation of the questionnaire was maintained between interviews, if the questionnaire was, in fact, not translated into the local language. That is, how can one be sure that the wording of the questions within the questionnaire was consistent from interview to interview?
5. Qualitative data collection and analysis: Owing to the multiple languages used and spoken throughout the data collection and analysis process, semantic validity checks and member-checks would be highly important. Were these performed? Please describe, if yes, in detail; and, if no, why not and highlight as a limitation.
6. Lines 126-131: To me, the lines about creating the dataset in Excel, then using StatTransfer to create a Stata® dataset, are unnecessarily detailed.
7. Line 133: Given the ranges of literacy in the study population, please describe (a) how informed consent was obtained, and (b) how participants signed these forms.
8. The methods should also detail the type of mixed methods design. An excellent (and open-access) publication on this topic is: Klassen AC, Creswell J, Plano Clark VL, Smith KC, Meissner HI.Best practices in mixed methods for quality of life research. Qual Life Res. 2012 Apr;21(3):377-80. doi: 10.1007/s11136-012-0122-x. Epub 2012 Feb 4.PMID: 22311251
9. The finding about perceptions of the causes of malaria is fascinating, and quite frankly, warrants very carefully planned knowledge translation. Please add a few words as to how you believe the findings need to be shared.
10. There are a few language, syntax and grammatical errors that warrant correction.
Author Response
Thank you for your comments! We will respond by referring to your points by black letters and responding by blue ones in following:
- Item 1. - not really. The reason for this was that the socio-demographic characteristics of population does not differ among settlements and they all were within a relatively small range
- Item 2. - no, we did not analyze results by monogamous/polygamous status of household
- item 3 - a major role in this is that of the community nurses/health advisors in field. Their respect in the community is high and lead to acceptance of research by households. On other end, as we discuss in limitations, this respect could lead to social desirability bias in results
- Item 4 - fully understand your concern and of course, full correctness of translation can hardly be expected. The way the field research team treated this was that same teams (researcher, local nurse, local interviewer) did all the interviews and visit.
- Item 5 - no, it wasn't and we added this into limitations, thank you
- Item 6 - point taken, but please allow us to keep it as it is. We believe it does not harm the manuscript
- Item 7 - there was a very detailed discussion on this issue within the project, within co-authors and in local setting. The decision was to collect oral consent due to illiteracy largely based on trust to local nurses and personnel.
- Item 8 - thank you for the reference!
- Item 9 - thank you for your positive comment! We added two sentences regarding this issue into Conclusions.
- Item 10 - we went through the text again and corrected, modified some issues.
Round 2
Reviewer 1 Report
- Good.
- I disagree with your response as that was not suggested to add it neither in your specific research part, nor in your results! Just as a literature discussion in your introduction to prove you have encircled your subject enough by different points of view based on other applications especially with the case of Ebola which I can not ignore if I was in your place. You could skip this remark as in the end you are free to discuss it or not.
- Good.
- Accepted. However, you may ask the editor to give you some additional time in order to present some simulations because the results are interesting, some data visualizations would be a plus!
- This does not deal with corona itself or your specific subject about children morbidity, but it has to do with people perceptions you brought up, so either your study was before or after, the problem would occur in the presence of any disease in that region. You could skip this remark as in the end you are free to discuss it or not.
In all cases. the paper deserves to be published in IJERPH, and I would even recommend it for citation after its publication.
Author Response
Dear Reviewer,
thank you for your understanding and comments. Here is our repsonse to remianing two issues from your comments:
item 2 - I added a sentence related to fear, misperceptions and lack of trust in relation to Ebola epidemic using one of references suggested by you; thank you!
item 4 - figures, charts.... - I replaced tables 1 and 3 by figures 1-4
Best regards
Gabriel Gulis on behalf of the author team
Reviewer 2 Report
Thanks. This draft looks and reads much better. Very good. Much more clear and readable. However, some minor editing still required. Examples are listed below. Overall, this is a great data set, good analysis and methodology, and will benefit academic science discourse (method, theory, data, etc.). Even more importantly, it will benefit applied fields (e.g., governments, NGOs, practitioners, researchers, health workers, local communities, etc.).
Some suggested edits (not exhaustive; rather representative/examples):
IMPORTANT NOTE: Probably best to use attached pdf file (when transferring to this platform; tables, indentations, italics, bolds, , center/left justifications, lines, numbering, etc. - a bit skewed - apologies).
Line 18: “…both quantitative and qualitative data contributions…” (might be a little more appropriate);
Line 92: “… of randomization. Households without children…”;
Line 100: “…50% of the households in order to…” (note: some other areas where commas are not necessary, or, authors may want to end sentences, break up larger compound complex sentences and make multiple, simpler, clearer, smaller sentences [increases readability and reduces confusion/ambiguity];
Lines 216-217: “The causes of child mortality in the villages relied solely on self-reported perception from 217 the mothers included in the survey.” [comma not necessary; may cause confusion].
Line 109: “…language, Temne. However, the questionnaire was performed…”
Line 122-125: “…The transcript aimed to be as true to the wording of the informants as possible. However, due to what is often referred to as African Pidgin English, where the phrasing is sometimes very different from standard English, statements were in some cases transcribed into a more formal language to make a more coherent text [14].
Lines 182-190:
“…Malaria is widely believed to be caused by oranges specifically, followed by sugary food; for example, candy, soda and honey. The belief is that oranges and sweet foods will make the blood glucose increase and that an imbalance in the body will cause malaria. Interestingly, this belief weighed much more heavily than the belief that mosquitoes transmit/cause malaria. However, for many, both causes were simultaneously acknowledged or believed.
When discussing diarrhoea, it was evident that informants believed many causes were related to nutrition. Even though contaminated drinking water was mentioned, it did not weigh heavily among informant perceptions. Rather, certain types of food were seen as the main cause, often in combination with breastmilk – “bad breast” was an often-mentioned as a primary cause of diarrhea. For further definition and clarification of “bad breast”, see textbox 3.”
Text box ‘bullets’ and alignments are still a bit confusing. I’m not sure if there are some program import/export issues (e.g., MS Word to another program/application), typos, or intentional. Suggested to clarify and straighten up a bit. See example below for textbox 2:
Note: “Defined from the semi structured interview and field notes: God’s will Not taking care of the child Witchcraft and evil spirits” removed from textbox as it seems to be reinserted in regular text above.
Textbox 2: Perceived causes for specific child diseases and morbidity based on semi structured interviews and field notes
|
Malaria
|
Diarrhea
|
Pneumonia
|
|
· Mosquitoes / mosquito bites |
· Contaminated drinking water |
· Exposing the child to a cold environment |
|
· Not sleeping under a bed net |
· Contaminated food |
· Exposing the child to a moist environment |
|
· Exposing the child to a cold environment |
· Unclean environment and improper use of latrines |
· Inhaling (contaminated) dust from the environment |
|
· Eating too many oranges |
· Having intercourse while still breastfeeding |
|
|
· Eating too many sugary foods: sodas and candy |
· Breastfeeding while having your period |
|
|
|
· Bad Breast; contaminated breastmilk |
|
|
|
· Some type of [real or perceived] nutrition problem: some beans, sweet potatoes, unripe mangoes, too many eggs, too many bananas |
|
Note: can also remove table gridlines.
|
Malaria
|
Diarrhea
|
Pneumonia
|
|
· Mosquitoes / mosquito bites |
· Contaminated drinking water |
· Exposing the child to a cold environment
|
|
· Not sleeping under a bed net |
· Contaminated food |
· Exposing the child to a moist environment
|
|
· Exposing the child to a cold environment |
· Unclean environment and improper use of latrines |
· Inhaling (contaminated) dust from the environment
|
|
· Eating too many oranges |
· Having intercourse while still breastfeeding
|
|
|
· Eating too many sugary foods: sodas and candy |
· Breastfeeding while having your period
|
|
|
|
· Bad Breast; contaminated breastmilk
|
|
|
|
· Some type of [real or perceived] nutrition problem: some beans, sweet potatoes, unripe mangoes, too many eggs, too many bananas |
|
Note: can also central justify is preferred by authors and/or journal editors:
|
Malaria |
Diarrhea |
Pneumonia |
|
· Mosquitoes / mosquito bites |
· Contaminated drinking water |
· Exposing the child to a cold environment
|
|
· Not sleeping under a bed net |
· Contaminated food |
· Exposing the child to a moist environment
|
|
· Exposing the child to a cold environment |
· Unclean environment and improper use of latrines |
· Inhaling (contaminated) dust from the environment
|
|
· Eating too many oranges |
· Having intercourse while still breastfeeding
|
|
|
· Eating too many sugary foods: sodas and candy |
· Breastfeeding while having your period
|
|
|
|
· Bad Breast; contaminated breastmilk
|
|
|
|
· Some type of [real or perceived] nutrition problem: some beans, sweet potatoes, unripe mangoes, too many eggs, too many bananas |
|
Lines 193-195: “Pneumonia was interesting because only three causes of pneumonia were mentioned. Perceived causes included exposing the child to a cold or moist environment and inhaling dust.
Textbox 4 provides some of the informant responses when asked how they identified supernatural causes of disease compared to biomedical causes.”
[note: new paragraph is the sentence is unrelated to pneumonia. May want to add a sentence or introductory claus: “Of further interest, Textbox 4 provides some of the informant responses when asked how they identified supernatural causes of disease compared to biomedical causes. It is also reemphasized that both supernatural and biomedical causes are often perceived a coexistent and equally valid among the informant population”].
Line 229: “…to take good care of them. While conducting the surveys, it…” [missing ‘surveys’ or ‘research’].
Lines 231-239: “… nutrition, adequate supervision and proper healthcare seeking behaviour. A separate study from Nigeria had similar results. The mothers believed that childhood pneumonia resulted partly from ‘carelessness of the mother’. This response is associated with similar responses in the current study; particularly, ‘not taking good care of the child’ (9). While investigating a more in-depth explanation of how and why certain aspects related to ‘taking good care’ affects the health of the child, informants explained that some of these (e.g., cold, special types of food, dirt etc.) “does not fit the system of the child” as if exposing the child to any of these things are found to somehow disturb a balance in the body and this will subsequently lead to disease. An example of this is the way oranges are perceived to disrupt the blood glucose and eventually lead to malaria. All of the mentioned aspects of ‘taking good care of a child’ are more or less related to physical aspects of human wellbeing.”
[note: several commas removed, some other minor edits, etc.].
Lines 240-303: Some editorial suggestions for remaining Discussion section:
In addition to perceived reasons for child mortality and morbidity, it became clearly apparent both from the survey and the interviews that biomedical reasons for disease and supernatural reasons coexist. This was evident in the survey where the participants were asked to name all the possible reasons they could think of for disease in children. Here, supernatural causes for disease such as witchcraft and evil spirits stand side by side with natural causes like contaminated drinking water, vector born malaria and lack of nutrients (poor or imbalanced nutrition). Hence, child health seems to a certain degree to be something that is influenceable by external factors. These factors can be affected by the caretaker’s behaviour towards the child related to ‘not taking good care of’ the child, and to some degree, something also affected by (or related to) supernatural or higher powers, such as witchcraft or God.
One study from Burkina Faso found a similar cultural phenomenon regarding the co-existence of beliefs, where mothers reported both to believe in malaria as being vector born disease transmitted through mosquitoes as well as caused by partially unrelated factors such as heavy oil consumption, work related fatigue, insufficient sleep, and exposure to sunlight. They also found that consumption of sweet fruits was seen as a cause for malaria [10]. Therefore, the perception that fruit can cause malaria is not something limited to Sierra Leone, nor West Africa. A study from Mozambique also found a connection between fruit consumption and malaria transmission, though the research team reported mosquito transmission to be the main believed cause of malaria [11].
The perception that breastmilk can be contaminated or dirty was equally prevalent in a study from South Africa where germs and contaminated drinking water as well as witchcraft was identified to cause diarrhoea [12].
This co-existence of both the biomedical [natural and scientific] and traditional [cultural beliefs and superstitions] paradigms seem to go hand in hand with the use of both facility-based medical treatment and herbal medicine from herbalists or traditionalists [local traditional medical experts, herbalists and healers]. The use of traditional medicine across Africa is quite substantial with up to 80% use across the region and is therefore not a phenomenon limited to Sierra Leone alone [15]. In our investigation, biomedical treatment was seen to be effective in several areas; for example, treating malaria, diarrhoea and pneumonia. The herbalist, however, was seen to be able to take care of the same illnesses, together with conditions perceived as being only caused by supernatural causes. Neurological symptoms such as convulsions, stiffness and severe fatigue were perceived to be rooted in supernatural causes alone. When asked who would best treat the child when sick from witchcraft, the agreed answer among the informants was the herbalist. In relation to identification of what was caused by witchcraft and what was a natural cause for disease, respondents noted that either elders or herbalists would be able to identify the nature of the cause. This might lead to health care seeking behaviour where the herbalist is sought before other modern healthcare specialists, services and facilities.
However, in the survey, only 0.3% of the informants stated that the first person they seek medical advice from was the herbalist. This seemed very inconsistent with the interviews and participant observations; and also differs with other studies in Sierra Leone on the use of herbal medicine where over half the respondents reported using herbal medicine (15). Some of the informants seemed to vary in their responses when it came to the use of herbal medicine. The reason there seemed to be a hesitancy talking about the use of herbal medicine and supernatural causes of disease might be due to the fact that the primary investigator, the translator, and the data collectors all represented the biomedical health system. This reluctance of informants to talk to health professionals about the use of herbal medicine also became evident when the informants were asked if they would tell the health care professional at the hospital about their use of traditional remedies. Five out of the five mothers said they defiantly would not. This tendency could potentially result in skewed responses and explain the low reporting to biomedical or modern medical specialists when it comes to use of herbalists in the survey.
One reason for seeking help at the herbalist first, was because of the system [Important Note to Authors here: … high costs and/or inaccessibility of the biomedical system…???; the power of the traditional cultural system…??? This sentence needs to explain what system and what aspect of the system]. The use of herbal remedies also seemed to be because informants claimed they see an effect from the herbs and therefore it makes sense for people to use them. All the informants, furthermore, stated to trust the herbalist and her/his knowledge. Furthermore, the health care worker specified in the interview, “they [the local population] trust them; that is the first point before seeking medical care.” Two studies from South Africa and Kenya, stated that traditional healers were used because they were trusted and respected in the communities as well, consistent with results in the current study. Another very central aspect when investigating the reasons for using the herbalist is the lower cost of herbal medicine and traditional remedies compared to a hospital or clinic visits. Bakshi et al. reported the same findings and noted a higher dependence in rural areas compared to urban [16].
In the current survey, 94.1% of the informants stated that the nurse or doctor would be the best person to help their child when ill. Several informants in the interview confirmed this. When the health care worker in the interview was asked if she believed that the hospital would be used more than herbalists if the people in the villages had the time, the money and the social support, she replied “they would go to the hospital, I am sure of that.”
Another trend noted in the current study, and found in other studies, is the fact that the local people often use the herbalist first and only seek formal modern medical attention if the condition does not improve or worsens despite the use of traditional remedies. Bakshia et al. found that people in Sierra Leone, especially in rural settings, only sought facility-based treatment for their children if the condition severely worsened [16, 17]. This can lead to severe delay in treatment and has potentially fatal consequences for children’s health and their likelihood for survival when faced with illness. Another concern regarding the safety of the children, is as expressed by the nutritionist and the nurse in charge of the paediatric ward, is the toxicity of the herbs used for treatment. Some traditional medicines and practices may has sever damaging effects, side effects or worsen problems. This remains unknown. Ranasinghe et al. points to the fact that the evidence on the area is limited and also inconsistent [15].
Lines 305-322: Some “limitations” editorial suggestions (note, moved the ‘Father’ statement in conclusion up to here – better location for the suggestion in this case):
One of the limitations of the study is the fact the primary investigator single-handedly performed the management and analysis of the data. However, as a positive tradeoff, this allowed for more consistent interpretation and coding of primary data as well as increased consistency in analysis and interpretation stages.
Secondly, because no accurate geographic and census data was available on how many households there were in what specific compound and no data on how many individuals were residents in the villages, a simple random sampling technique was not possible. Furthermore, due to the nature of the survey and the initial purpose of the data collection (namely, to investigate child health in relation to the Community Child Health project in Masanga 311 Outreach), a cluster sampling method was also not appropriate. Thus, a randomised collection was not achievable, and a convenience sampling was deemed necessary. It was decided an oversampling of the calculated sample size of 254 households would compensate (i.e., if representativeness was not secured through a random sampling process, at least a larger sample set of 50% of target population would be equally in not more reflective and accurate). The target for minimum sample size was therefore raised to be 50% of all the households.
Thirdly, an information bias that was suspected in the study was social desirability response bias. This is suspected because of the observed inconsistency between the survey answers and the interviews. If only taking the interview responses and participant observations into consideration, the survey seems to be a more positive representation of life for children in the villages. Therefore, there is a risk that the survey method alone underrepresents or diminishes the actual challenges and barriers the informant communities face.
Fourthly, another limitation is language; especially in cases concerning qualitative interviews where lack of detailed translation/re-translation and therefore detailed semantic analysis can lead to bias. However, the use of spoken interviews and surveys using local languages and face to fact interaction helped compensate for language difference biases.
Fifthly, only females and mothers were the primary target population for data collection. It might be important to include father and extended family responses in future studies (aunts, uncles, cousins, grandparents, and even caregiver friends).
Lastly, other limitations, as usual, include time, funding, staff, accessibility and other support constraints normative of many research endeavors in most fields. Nevertheless, with a sample set of 50% of the population and 100% response rate, these normative constraints were adequately compensated for – especially thanks to the positive support of local communities, leaders, and healthcare personnel and institutions.
Lines 324-341: Conclusion: Need a bit of intro (many readers will skip to conclusion first) and a little more of the issues discussed in discussion. See/use examples below:
A mixed-method research project was conducted in order to assess perceived healthcare problems, causes, and treatments for children 5 years old and younger among impoverished gold mining and farming villages in the Tonkolili district of Sierra Leone. 50% of households were samples with 100% response rate. The main perceived cause for ill health among children five years old and younger (five and under), was due to lack of care, expressed as “not taking good care of the child”. This was related to poor hygeine practises, unsafe environment, insufficient nutrition, inadequate supervision and poor healthcare seeking behaviour. The perception of ill health in children was to a high degree associated with the parent’s ability to cater for the child’s physical needs. A more in-depth explanation of why and how these aspects affect the child was believed to be because these factors could upset the system of the child – they were seen to have a kind of inner balance. In relation to perceived causes of child mortality, biomedical causes for disease such as malaria, pneumonia and diarrhoea together with supernatural causes for disease such as witchcraft, evil spirits and God’s will, were seen to coexist. A clearly defined line between the two did not seem to exist and the causes of disease where seen to be caused either because the children were not taken care of (more related to biomedical factors and parental healthcare of their children), and/or, because of supernatural causes. Children’s ill health and mortality was thereby seen both as an element that could be influenced by the parent and their actions (or lack thereof), and simultaneously something that was out of the caretaker’s hands and up to other people such as those who are able to bewitch, cast spells, etc.; or, up to the will of God.
One of key findings of the study is that of traditional perceived causes of malaria as being related to diet and other factors not directly related to the actual parasite and mosquito transmission – again demonstrating how important and strong the traditional beliefs of local population are. This also relates to the often preferred use of local herbalists and medical healers instead of modern specialists, clinics and other facilities. The preferred use of herbalists and elders first in many cases may be economic. Clinics and modern treatment are more expensive. However, it also may be related to traditional beliefs and practices as well as other socio-cultural aspects such as trust. Locals trust and respect the local elders, healers and herbalists. This is powerful and influential in regards to parental choices and actions. In many cases, only if conditions for the children worsen, professional modern treatment is sought. By then, successful treatment may be too late. These issues need to be taken in account while planning future research, health literacy programs, disease prevention practices, disease treatment practices, treatment options, and overall healthcare planning in Sierra Leone and similar settings.

Author Response
Dear Reviewer,
thank you for all your comments and suggestions to edit the text! I accepted all your changes; in some cases even added more editing changes. Special thanks for attaching your review file, which allowed me to respond to your comments directly in it: please see it uploaded. Of course, I edited the text of the manuscript by track changes, so you can see all changes also there clearly
Best regards
Gabriel Gulis on behalf of author team
